# ODELAM, rapid sequence-independent detection of drug resistance in isolates of *Mycobacterium tuberculosis*

Thurston Herricks[1†], Magdalena Donczew[2†], Fred D Mast[1], Tige Rustad[1], Robert Morrison[1], Timothy R Sterling[3], David R Sherman[2*], John D Aitchison[1,4*]

[1]Center for Global Infectious Disease Research, Seattle Children's Research Institute, Seattle, United States; [2]Department of Microbiology, University of Washington, Seattle, United States; [3]Division of Infectious Disease, Department of Medicine, Vanderbilt University Medical Center, Nashville, United States; [4]Department of Pediatrics, University of Washington, Seattle, United States

**Abstract** Antimicrobial-resistant *Mycobacterium tuberculosis* (Mtb) causes over 200,000 deaths each year. Current assays of antimicrobial resistance need knowledge of mutations that confer drug resistance, or long periods of culture time to test growth under drug pressure. We present ODELAM (One-cell Doubling Evaluation of Living Arrays of Mycobacterium), a time-lapse microscopy-based method that observes individual cells growing into microcolonies. ODELAM enables rapid quantitative measures of growth kinetics in as little as 30 hrs under a wide variety of environmental conditions. We demonstrate ODELAM's utility by identifying ofloxacin resistance in cultured clinical isolates of Mtb and benchmark its performance with standard minimum inhibitory concentration (MIC) assays. ODELAM identified ofloxacin heteroresistance and the presence of drug resistant colony forming units (CFUs) at 1 per 1000 CFUs in as little as 48 hrs. ODELAM is a powerful new tool that can rapidly evaluate Mtb drug resistance in a laboratory setting.

**\*For correspondence:**
dsherman@uw.edu (DRS);
John.Aitchison@seattlechildrens.
org (JDA)

[†]These authors contributed equally to this work

**Competing interests:** The authors declare that no competing interests exist.

## Introduction

The continued and accelerated emergence of anti-microbial resistance is a global threat and antimicrobial-resistant infections caused by *Mycobacterium tuberculosis* (Mtb) are particularly concerning. In 2018, approximately 484,000 people developed multidrug-resistant tuberculosis and an estimated 214,000 people died from rifampicin-resistant or multidrug-resistant tuberculosis (Geneva: *World Health Organization, 2019*). Approximately 10% of all multidrug resistant tuberculosis infections are resistant to at least four commonly used anti-TB drugs (*Dorman et al., 2018*).

Rapid, reliable diagnosis is a key to the prevention and effective treatment of antimicrobial resistant infections (*Boolchandani et al., 2019*). In the case of Mtb, culture-based methods such as common agar plates or more advanced indicator tubes with fluorescent growth reporters are gold standards for identifying sensitivity to any antibiotic, but these testing methods require one to four weeks to produce results (*Kim, 2005*; *Lawson et al., 2013*; *Kontos et al., 2004*; *Garrigó et al., 2007*). Biomarker methods, such as the PCR-based Gene-Xpert assay, can rapidly identify antimicrobial resistant Mtb infections but require knowledge of specific genetic markers associated with resistance and are thus insensitive to unknown resistance mechanisms (*Dorman et al., 2018*). Detecting antimicrobial-resistant Mtb infections is especially challenging when bacterial sub-populations are present that possess differing levels of antimicrobial sensitivity (*El-Halfawy and Valvano, 2015*). If present in relatively low abundance, these heteroresistant populations may evade detection until after treatment begins. Failure to rapidly identify and treat an antimicrobial-resistant or

heteroresistant Mtb infection contributes to disease progression, treatment failure and tuberculosis relapse (*Shin et al., 2018*).

We developed ODELAM (One-cell Doubling Evaluation of Living Arrays of *Mycobacterium*), a time-lapse microscopy-based method designed to quantify growth phenotypes of populations of individual Mtb cells and colony forming units. Generally imaging platforms used to study mycobacteria use either microfluidic techniques or growth on solid media. Microfluidic platforms provide high resolution phenotypic analysis of mycobacterial cell biology but are limited in the number of cells that can be readily observed (*Aldridge et al., 2012*; *Golchin et al., 2012*; *Wakamoto et al., 2013*). Conventional assays on solid media provide crude growth analysis, including antibiotic sensitivity, on large populations of cells but require up to 8 weeks for observations whereas specialized microfluidic methods require approximately 9 days (*Choi et al., 2014*; *Barr et al., 2016*).

ODELAM bridges these disparate methods by assaying growth temporally at a mesoscale, on up to ~100,000 colony forming units (CFUs) in a single experiment. Originally designed for yeast, the adaptation offers improvements on phenotypic analysis and diagnostics for detecting and characterizing populations of drug resistant Mtb (*Herricks et al., 2017*). While commercial time-lapse and laboratory based genomic screening tools are available (*Lee et al., 2019*; *Baryshnikova et al., 2010*; *Bean et al., 2014*; *Zackrisson et al., 2016*; *Golchin et al., 2012*), ODELAM uniquely enables rapid quantitative measurements of growth kinetics of homogeneous and heterogeneous cultured clinical isolates of Mtb in as little as 30 hrs and under a variety of environmental conditions, including antimicrobial drug pressures.

We demonstrate ODELAM by analyzing ofloxacin (OFX) resistance in cultured Mtb clinical isolates. ODELAM was first benchmarked with the laboratory strain H37Rv and then two cultured clinical isolates were tested, an OFX-resistant isolate and an isolate with OFX heteroresistance (*Eilertson et al., 2016*). Microscopy-based kinetic growth analysis of these strains under increasing drug pressures revealed characteristic growth and drug sensitivity phenotypes of each strain and specifically demonstrate the ability to rapidly detect heteroresistant sub-populations in a cultured clinical isolate. ODELAM is a powerful new tool that can enable the timely identification of Mtb antimicrobial resistant phenotypes in a laboratory setting.

## Results

ODELAM is engineered to quantitatively assess growth kinetics of large populations of individual Mtb CFUs. In each region of interest, ODELAM can observe up to 1500 CFUs and quantify their morphological and multiparameter growth phenotypes. A typical experiment involves 80 or 96 regions of interest. Single bacilli and small clusters containing two or more bacilli are observed over time as they grow into microscopically observable colonies (*Figure 1A*). In these experiments, Mtb bacilli were observed to grow immediately after spotting on media. Mtb CFUs were recorded for a total of 96 hrs and growth curves were extracted from the projected area of the individual colonies over time and fitted to the Gompertz function (*Figure 1B*; *Herricks et al., 2017*). The kinetic parameters doubling time (Td), lag time (T-Lag), time in exponential phase (T-Exp) and number of doublings (Num Dbl) were derived from the Gompertz function and plotted as normalized frequency histograms which revealed population distributions (*Figure 1C*). Summary non-parametric statistics for the isolates measured are presented as interquartile range (IQR) with respect to the median (*Table 1*). For example, H37Rv, an Mtb strain commonly used in research laboratories, had a median lag time of 0.5 s and grew with a doubling time median of 22.9 hrs (IQR 7.8). H37Rv bacilli grew in exponential phase for a median of 52.5 hrs (IQR 33.8) and during this time underwent a median of 3.1 doublings (IQR 1.6). These data are consistent with bulk laboratory measurements (*Peñuelas-Urquides et al., 2013*) and demonstrate the innate heterogeneity of the genetically homogeneous clonal population.

### Growth phenotype of Mtb strain H37Rv exposed to ofloxacin

One of the main advantages of ODELAM compared to standard bulk culture experiments is its ability to assess population responses on a single-CFU level. We therefore used ODELAM to reveal growth parameter values for individual CFUs in response to increasing drug pressures and compared these results to standard dose-response measurements (*Figure 2*). We first focused on H37Rv. H37Rv is sensitive to ofloxacin with minimum inhibitory concentration (MIC) of 0.5 µg/ml as

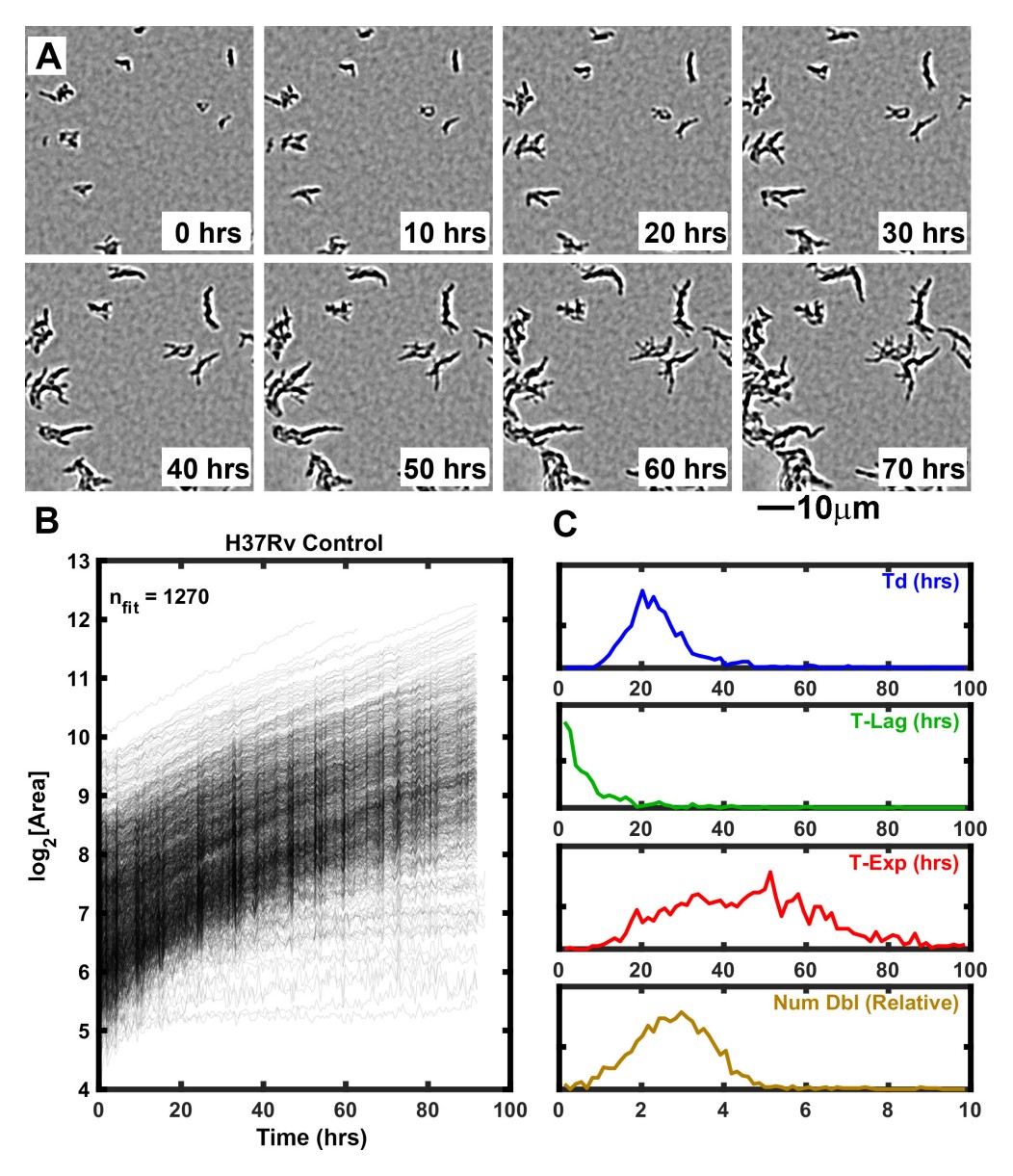

**Figure 1.** Growth of Mtb strain H37Rv into microcolonies over time observed with ODELAM. (**A**) Selected time-lapse snapshots of Mtb colonies growing on a solid medium. Mtb CFUs are shown growing on a 60 µm x 60 µm region of 7H9-GO agar over 96 hrs time course. (**B**) Growth curves for a total of 1276 colonies were recorded from a 1.2 mm x 0.9 mm region over 96 hrs and plotted. (**C**) Population histograms of the all extracted growth parameters (Doubling Time, Lag Time, Exponential Time and Number of doublings).

measured by normalized population growth in a standard dose-response curve (*Figure 2B*). By ODE-LAM, we observed H37Rv to initially grow at all concentrations of OFX indicating that OFX's toxicity does not alter growth kinetics over the first 24 hrs (*Figure 2A*). This is consistent with OFX's mechanism of action on the enzyme target, DNA gyrase, which leads to the accumulation of DNA damage (*Manjunatha et al., 2002*; *Gore et al., 2006*; *Willmott et al., 1994*). Above the MIC, toxicity led to cessation of growth, detected as an exit from exponential phase. This is reported as and a reduction in time spent in exponential phase (T-Exp) and in the number of doublings (Num Dbl) as shown by shifts of the population histograms (*Figure 2C*). Narrowing of the population histograms reflects a more uniform population. The population distribution at the MIC remained relatively broad but was narrower than at lower drug concentrations or in controls. Notably, the doubling time distributions

**Table 1.** Summary growth statistics of isolates.

| Condition tested | CFUs total | Num of rep | Med td (hrs) | IQR td (hrs) | Med T-Lag time (hrs) | IQR T-Lag time (hrs) | Med T-Exp (hrs) | IQR T-Exp (hrs) | Med Num Dbl | IQR Num Dbl |
|---|---|---|---|---|---|---|---|---|---|---|
| Control-H37Rv | 5413 | 8 | 22.9 | 7.8 | 0.0 | 3.7 | 52.5 | 33.8 | 3.1 | 1.6 |
| 0.0625 µg/ml-H37Rv | 3380 | 4 | 23.4 | 9.6 | 0.0 | 2.8 | 51.8 | 28.1 | 3.0 | 1.5 |
| 0.125 µg/ml-H37Rv | 2946 | 4 | 24.3 | 10.0 | 0.0 | 2.9 | 53.0 | 29.5 | 3.0 | 1.4 |
| 0.5 µg/ml-H37Rv | 2231 | 4 | 22.6 | 10.7 | 0.9 | 6.2 | 35.2 | 16.2 | 2.0 | 0.9 |
| 2 µg/ml-H37Rv | 5354 | 8 | 19.2 | 13.7 | 3.9 | 6.1 | 17.3 | 8.2 | 0.9 | 0.6 |
| 4 µg/ml-H37Rv | 1412 | 4 | 21.1 | 12.5 | 3.9 | 5.2 | 14.7 | 7.7 | 0.6 | 0.4 |
| 8 µg/ml-H37Rv | 1728 | 4 | 20.8 | 12.1 | 2.8 | 4.0 | 11.5 | 6.5 | 0.5 | 0.3 |
| 16 µg/ml-H37Rv | 894 | 4 | 22.9 | 24.4 | 1.5 | 4.4 | 11.2 | 12.4 | 0.5 | 0.3 |
| Control-TRS10 | 3449 | 8 | 28.4 | 8.5 | 2.8 | 9.3 | 79.7 | 54.2 | 3.5 | 2.5 |
| 0.0625 µg/ml-TRS10 | 2181 | 4 | 28.2 | 6.9 | 1.9 | 8.1 | 81.7 | 55.6 | 3.8 | 2.6 |
| 0.125 µg/ml-TRS10 | 2014 | 4 | 28.5 | 7.6 | 0.0 | 5.4 | 80.3 | 61.2 | 3.7 | 2.8 |
| 0.5 µg/ml-TRS10 | 1167 | 4 | 28.3 | 6.3 | 0.0 | 6.0 | 81.8 | 62.5 | 3.8 | 3.1 |
| 2 µg/ml-TRS10 | 2982 | 8 | 30.6 | 17.9 | 2.3 | 9.2 | 73.7 | 42.6 | 2.8 | 2.2 |
| 4 µg/ml-TRS10 | 1190 | 4 | 53.1 | 26.0 | 0.0 | 7.4 | 65.7 | 38.7 | 1.5 | 1.1 |
| 8 µg/ml-TRS10 | 1063 | 4 | 57.8 | 28.4 | 0.0 | 2.1 | 56.7 | 27.4 | 1.3 | 0.7 |
| 16 µg/ml-TRS10 | 966 | 4 | 46.0 | 21.8 | 0.0 | 13.8 | 61.8 | 44.8 | 1.6 | 1.3 |
| Control-ADB42 | 5599 | 8 | 24.4 | 7.8 | 1.1 | 8.2 | 64.3 | 42.9 | 3.4 | 2.0 |
| 0.0625 µg/ml-ADB42 | 3762 | 4 | 24.7 | 7.2 | 0.0 | 5.2 | 63.8 | 35.7 | 3.4 | 1.8 |
| 0.125 µg/ml-ADB42 | 3980 | 4 | 25.3 | 7.4 | 0.0 | 4.8 | 63.8 | 35.1 | 3.4 | 1.7 |
| 0.5 µg/ml-ADB42 | 3121 | 4 | 25.0 | 9.7 | 0.0 | 6.2 | 40.6 | 31.7 | 2.0 | 1.4 |
| 2 µg/ml-ADB42 | 4995 | 8 | 26.9 | 18.0 | 1.9 | 8.0 | 29.5 | 43.3 | 1.1 | 1.7 |
| 4 µg/ml-ADB42 | 1440 | 4 | 36.7 | 27.7 | 5.0 | 13.9 | 38.8 | 51.6 | 0.9 | 1.9 |
| 8 µg/ml-ADB42 | 1287 | 4 | 54.7 | 56.2 | 4.1 | 13.2 | 40.0 | 39.2 | 0.6 | 0.9 |
| 16 µg/ml-ADB42 | 910 | 4 | 17.5 | 34.8 | 31.5 | 32.6 | 46.6 | 36.4 | 0.6 | 1.0 |

did not appreciably change; cells maintained similar doubling rates prior to growth cessation. Overall, the primary effect of OFX as detected by ODELAM was to arrest H37Rv growth after 2–3 doublings at the MIC and by the first doubling at high drug concentrations, while not appreciably affecting the rate at which H37Rv doubles prior to growth arrest (*Table 1*).

## Growth phenotype of a resistant strain TRS10 exposed to ofloxacin

We observed a cultured clinical isolate TRS10, which contains a missense mutation in the gene encoding gyrase A (D94G) that increases the isolate's MIC to OFX (*Eilertson et al., 2016*; *Hooper, 2001*). In the absence of drug pressure, the median doubling time of TRS10 was approximately 20% longer than that of H37Rv (*Table 1*). The MIC of TRS10 to OFX was 16 µg/ml as measured by the normalized population growth in a standard dose-response curve (*Figure 3B*). By contrast to H37Rv, which ceased to grow at its MIC, TRS10 slowed its doubling time as OFX concentration increased up to, and including, its MIC (*Figure 3A*). Further, TRS10's doubling time slowed dramatically from ~30 to~50 hrs as drug concentration shifted from 2 to 4 µg/ml (*Table 1*).

Total growth (Num Dbl) is influenced by all three growth parameters (Td, T-Lag and T-Exp). Therefore, to discern the contribution of these parameters to total growth we determined the

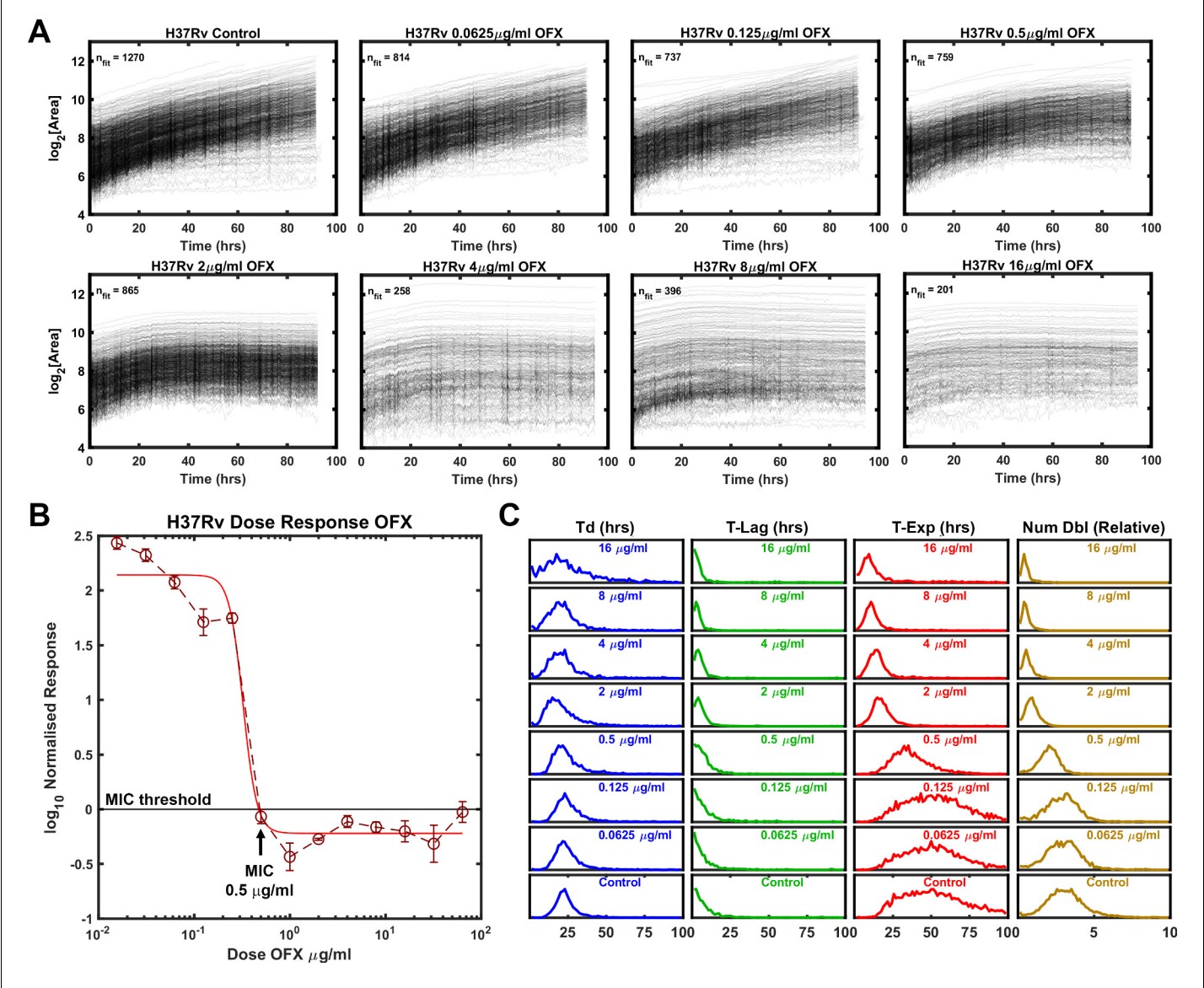

**Figure 2.** Growth of Mtb strain H37Rv during exposure to ofloxacin. (**A**) Growth curves of H37Rv under increasing OFX drug concentrations measured by ODELAM. The flattening of the curves corresponds to the MIC of about 0.5 μg/ml OFX and corresponds to the batch culture dose response (**B**). (**C**) Population histograms of the growth parameters showing that the time in exponential growth (T-Exp) is reduced as OFX concentration increases, which in turn leads to a reduced number of doublings.

population effect size, which quantifies the magnitude of effect for each parameter (Td, T-Lag, T-Exp, Num Dbl). In this case, the Kolmogorov-Smirnov (K-S) statistic was used as a score to evaluate the effect size and plotted against the concentration of OFX for each strain (*Figure 4*; *Massey, 1951*). As a non-parametric test, the K-S statistic allows the comparison of the distributions without presumption of their shape or parameters that underlie them. For H37Rv, the effect size for time in exponential phase tracked with the number of doublings as the concentration of OFX increased above 0.5 μg/ml while doubling time and lag time effect sizes did not (*Figure 4A*). In contrast, for TRS10 the effect size for doubling time tracked with the number of doublings as the concentration of OFX increased (*Figure 4B*). This OFX effect on TRS10's doubling time as opposed to the cessation of growth observed for H37Rv (T-Exp) likely reflects the reduced binding of OFX to the mutant gyrase (*Maruri et al., 2012*; *Willmott et al., 1994*). Together these results show the ability of ODELAM and this analysis to reveal strain-dependent responses to antibiotics that would be

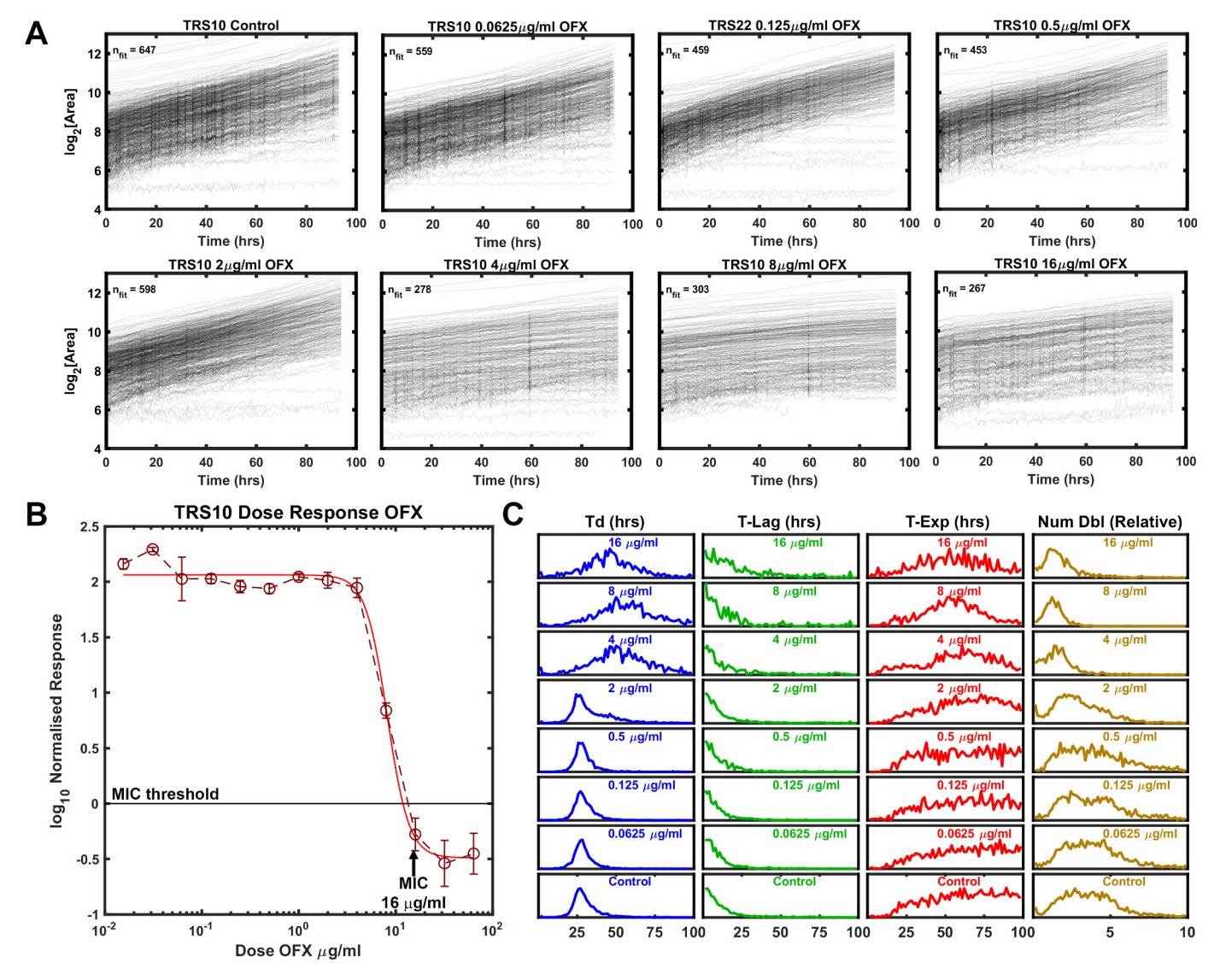

**Figure 3.** Growth of Mtb strain TRS10 during exposure to ofloxacin. (**A**) Growth of clinical isolate TRS10 appears to slow under increasing OFX pressure. (**B**) Batch culture dose response curve for TRS10, indicating an MIC of 16 µg/ml OFX. (**C**) ODELAM histograms with increasing OFX concentration, showing that up to the MIC, doubling time appears to increase while the time in exponential phase is less affected.

missed by standard MIC assays. The examples of H37Rv and TRS10 responding to OFX (*Figures 2* and *3*), as assayed by ODELAM, discerns these differing growth phenotypes and can inform on potential mechanisms of drug action.

## Detecting heteroresistance

Heteroresistance in cultured clinical isolates is prevalent and confounds diagnosis and treatment of Mtb (*Sebastian et al., 2017*). Therefore, we tested the ability of ODELAM to rapidly detect heteroresistance. TRS10 was mixed with H37Rv to generate strain ratios ranging from 1:1 to 1:4166. Mixed cultures were assayed by standard dose-response batch culture MIC assay (*Figure 5A*) and with ODELAM (*Figure 5B and C*) at a concentration of 2 µg/ml OFX which discriminates OFX-sensitive and OFX-resistant CFUs.

To evaluate the sensitivity of the MIC assay to heteroresistant cultures, the MICs of the dilution series were plotted (*Figure 5A*). The observed MIC varied across the dilution series, increasing with the proportion of resistant cells. While the assay is sufficiently sensitive to detect shifts in the MIC at

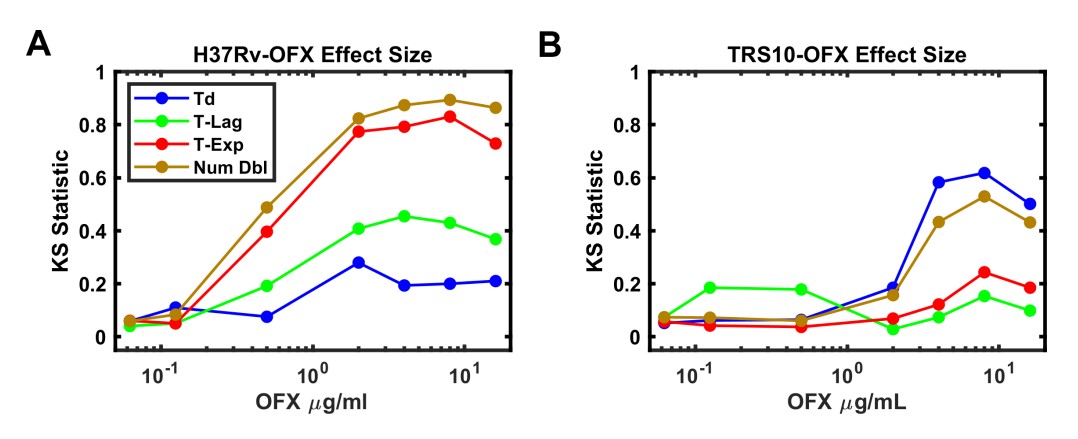

**Figure 4.** Effect Size of OFX on growth parameters. (**A**) Effect size as measured by the Kolmogorov-Smirnov statistic vs OFX concentrations for H37Rv. Note, the number of doublings observed tracks with the time growing exponentially (T-Exp). (**B**) Effect size as measured by the Kolmogorov-Smirnov statistic vs OFX concentrations for TRS10. Note, doubling time (Td) tracks with number of doublings (**B**).

very low relative concentrations of resistant cells, it does not distinguish between a monoculture with a specific resistance phenotype and a mixed population with heteroresistance. Thus, the standard dose-response batch culture MIC assay yielded inaccurate results and was unable to reflect the MIC of the constituent population.

By contrast ODELAM directly observed OFX resistant CFUs in the mixed culture (*Figure 5B,C*). Resistant and sensitive CFUs were distinguished by their number of doublings and their growth in exponential phase. In this ODELAM assay, a single OFX resistant CFU was detected from the 0.04% TRS10 mixture (*Figure 5C*). Accordingly, there was an increase in the proportion of OFX-resistant CFUs detected as the percentage of TRS10 increased (*Figure 5B,C*). The 50% mix detected 200 resistant CFUs out a total of 721 CFUs. Similarly, 28 of 970 CFUs were detected in the 6.25% TRS10 mix, and 12 of 1144 were detected in the 1.56% TRS10 mix (*Figure 5C*). A summary of the results from 12 dilution mixes and 8 replicates are plotted (*Figure 5B*). The relationship between dilution and detection was linear from 50% to 0.2%. Below 0.2% the expected number of TRS10 CFUs fall to around 1 to 5 in 1000 individuals and because each region of interest observes about 1000 CFUs, the measurements become dominated by chance. ODELAM consistently observed fewer resistant TRS10 CFUs from the mixture than expected from the $OD_{600}$ measurements used to generate the mixture (*Figure 5B*) and likely reflects the vagaries associated with using $OD_{600}$ to predict CFUs, which is known to vary between strains (*Peñuelas-Urquides et al., 2013*).

## Detecting heteroresistance in cultured clinical isolate ADB42

Clinical isolate ADB42 was previously observed to be OFX resistant with MIC ranging between 8 and 256 μg/ml of OFX (*Eilertson et al., 2016*; *Eilertson et al., 2014*). By whole genome sequence analysis, the canonical OFX resistance conferring gyrase-A D94G SNP was not present at a frequency sufficient to be associated with the observed OFX resistance; other mechanisms were attributed to the observed phenotype (*Eilertson et al., 2016*). In our hands, the standard dose-response assay yielded a MIC of 16 μg/ml (*Figure 6A*). By ODELAM population measurements, the histograms revealed features indicative of phenotypic heterogeneity. Prominent tails were observed on the right-hand side of the exponential phase and number of doublings population distributions for OFX concentrations at and above 0.5 μg/ml (*Figure 6B*), suggesting the presence of at least two populations. These populations could be segregated by a simple threshold of 2 doublings as detected by ODELAM on 2 μg/ml OFX (*Figure 7*). The CFU growth curves corresponding to these populations showed either OFX sensitivity (*Figure 7A*) or OFX resistance (*Figure 7B*) similar to those observed for the experimentally mixed populations of H37Rv and TRS10 (*Figure 5C*).

ADB42 was grown in culture and subjected to transcriptome analysis by RNA-sequencing, which revealed the presence of a gyrase-A D94G SNP at a frequency of 15% (*Figure 8A*). Across the

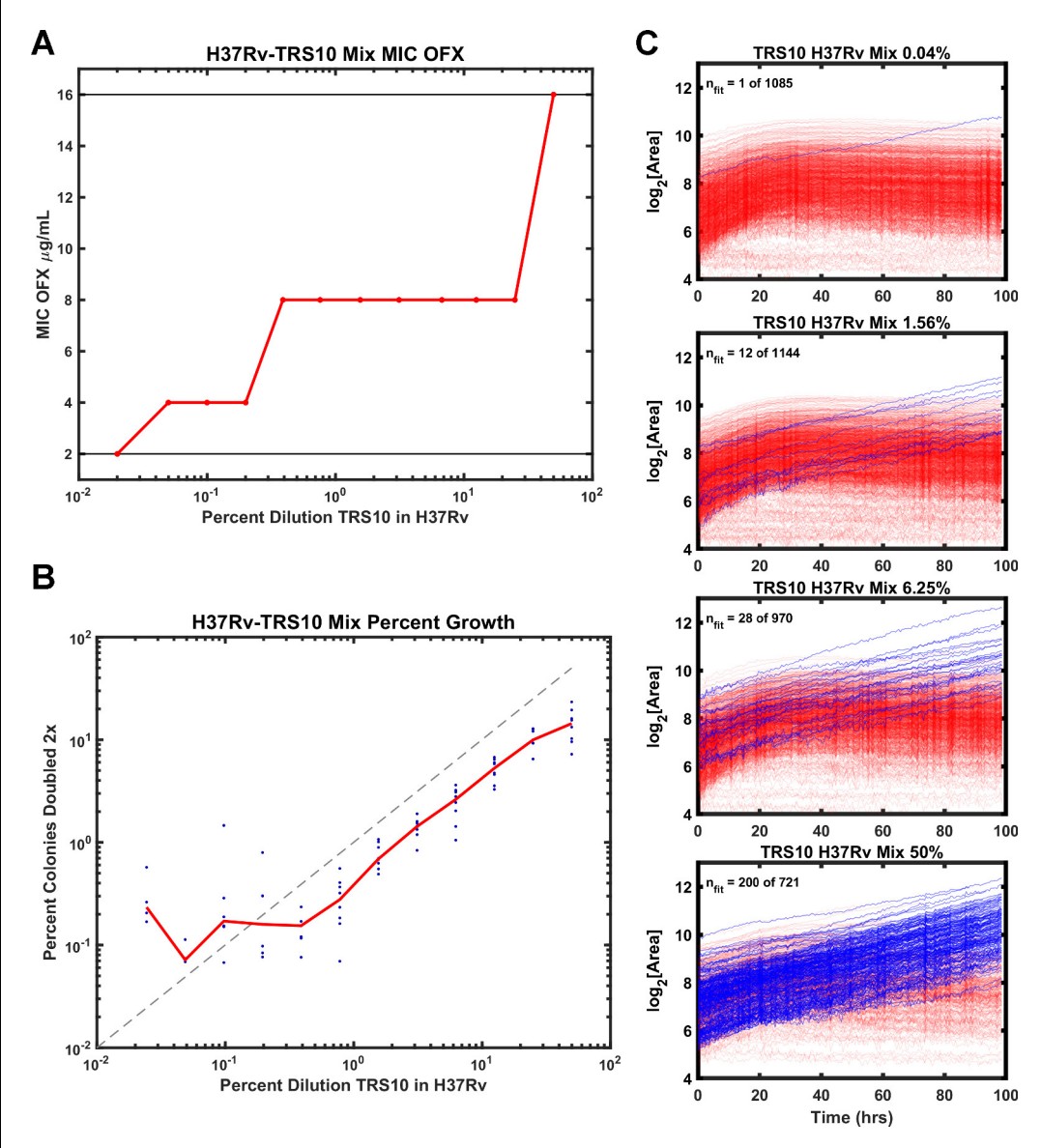

**Figure 5.** Comparison of ODELAM and MIC assay in detecting heteroresistance. (A) Dilution of TRS10 into H37Rv and the corresponding MICs obtained from a bulk culture assay. (B) Comparison of the percentage of TRS10 in H37Rv generated by dilution against the percentage of CFUs observed to grow more than 2× by ODELAM. The blue dots are replicate measurements from a given dilution and the red line is the mean of each replicate. (C) ODELAM growth curves for TRS10/H37Rv mixtures. Blue traces are resistant and red traces are sensitive to OFX.

experiments reported here, ODELAM measured roughly 24% of the ADB42 CFUs were resistant to OFX, which is within a 95% confidence interval of random sampling modeled by a binomial distribution (*Table 2*). Ninety-six individual clones of ADB42 were isolated and their growth on 2 μg/ml OFX was measured in a spot assay (*Figure 8B and C*). These clones were also measured with ODELAM to evaluate their growth kinetics and OFX resistance (*Figure 9*). Thirty-three clones exhibited OFX resistance at 2 μg/ml OFX in both assays. In contrast to the parental cultured clinical isolate, these individual clones did not display phenotypic heterogeneity by ODELAM (data not shown). In total, these data indicate that cultured clinical isolate ADB42 contains two distinct populations one of which is resistant to ofloxacin and likely contains a D94G gyrase-A mutation that confers OFX resistance.

In further support of these findings, we sequenced the quinolone-resistance-determining region (QRDR) of *gyrA* in fourteen of these ADB42 clones, 10 OFX-resistant and four OFX-sensitive. All 10

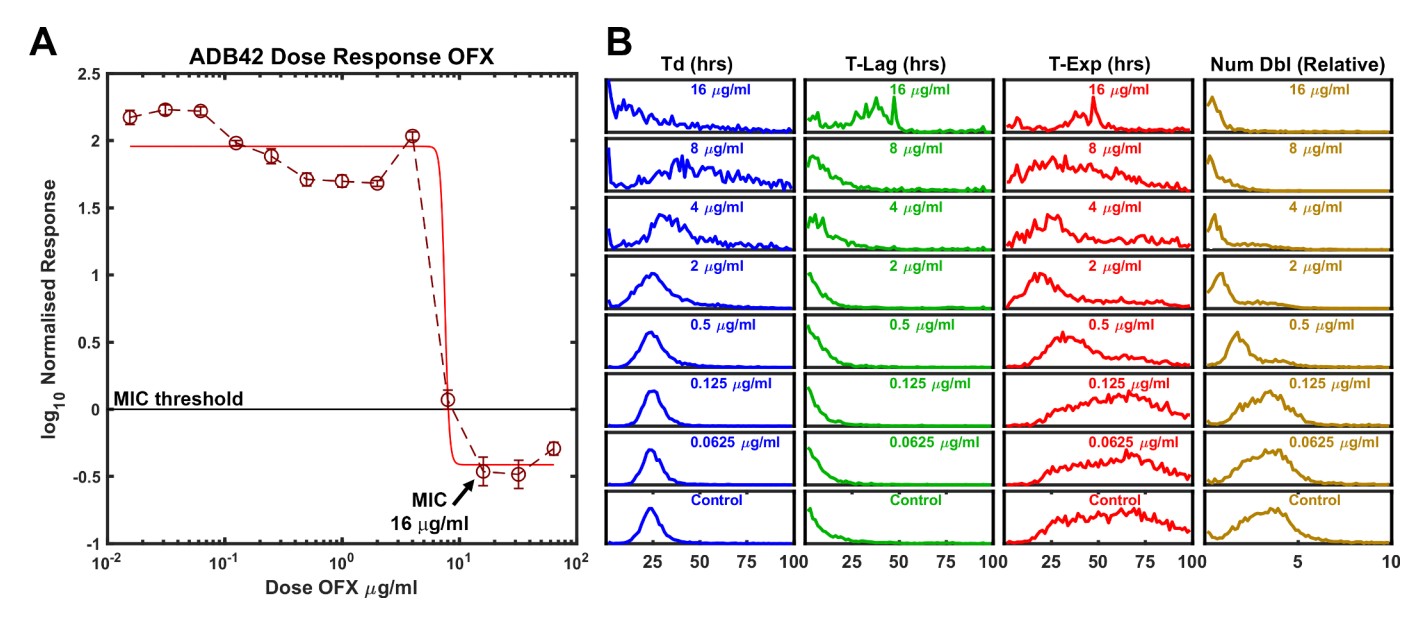

**Figure 6.** ODELAM analysis of Mtb strain ADB42 reveals heteroresistance to OFX. (**A**) Batch culture-based dose response curve of ADB42 showing an OFX MIC of about 16 µg/ml. (**B**) At 2 µg/ml OFX, tails in the doubling time, T-Exp, and number of doublings indicate that a second population may be present.

resistant isolates contained a SNP A7582G in *gyrA*, which corresponds to the canonical D94G mutations that confers OFX resistance (*Willmott et al., 1994*). The four sensitive clones did not contain this SNP (*Table 3*). Additional gyrase mutations were present in the QRDR of *gyrA* but these mutations were not associated with OFX sensitivity or resistance (*Table 3*).

## Observation of OFX induced cell lysis

ODELAM directly observes cells and therefore informs on additional cellular phenotypes. After Mtb growth arrest on OFX, some cells appear to lyse as detected by a change in contrast (*Figure 10A*), which we interpret as a consequence of cytotoxicity (*Figure 10A*). This phenotype is observed as a sharp reduction in a colony's size in its growth curve (*Figure 10B*). The percentage of colonies that have this phenotype increased with OFX concentration and became apparent at concentrations greater than 0.5 µg/ml OFX. The frequency of lysis scales with the sensitivity of the strain to OFX. Here H37Rv showed as much as 95% of the tracked colonies exhibiting cell lysis, where as ADB42 and TRS10 showed 65% and 30%, respectively (*Figure 10C*). These observations by ODELAM highlight the complex transition from a cytostatic to a cytolytic phase of the drug-organism interaction.

## Rate of observation time convergence

Rapid detection of drug sensitivity improves in the successful treatment of tuberculosis. To establish the minimum time required by ODELAM to reliably identify anti-microbial resistance and sensitivity, we determined the length of time necessary to estimate a colony's set of 4 kinetic growth parameters. Each growth curve is fit using an algorithm optimized for ODELAM data. Ideal growth curves were generated in silico varying the kinetic parameters with random noise added. The Gompertz fit algorithm then estimated growth parameters for each noisy growth curve (*Figure 11A–C*). To simulate observing growth data over time, an increasing number of time points were included into the fit algorithm. As more time points from simulated growth curves were added, we evaluated if the estimated parameters converged to stable values (*Figure 12A*). Stable estimates were defined by how many time points were needed to be included for the differences in sequential estimates to differ by a precision of less than 2.5% or 5% (*Figure 12B–C*). Using this routine, we calculated that the time required to reliably determine the set of kinetic parameters for a growth curve must be ~1.5 times longer than the time needed for a CFU to exit exponential phase. In the case of H37Rv, which, in the

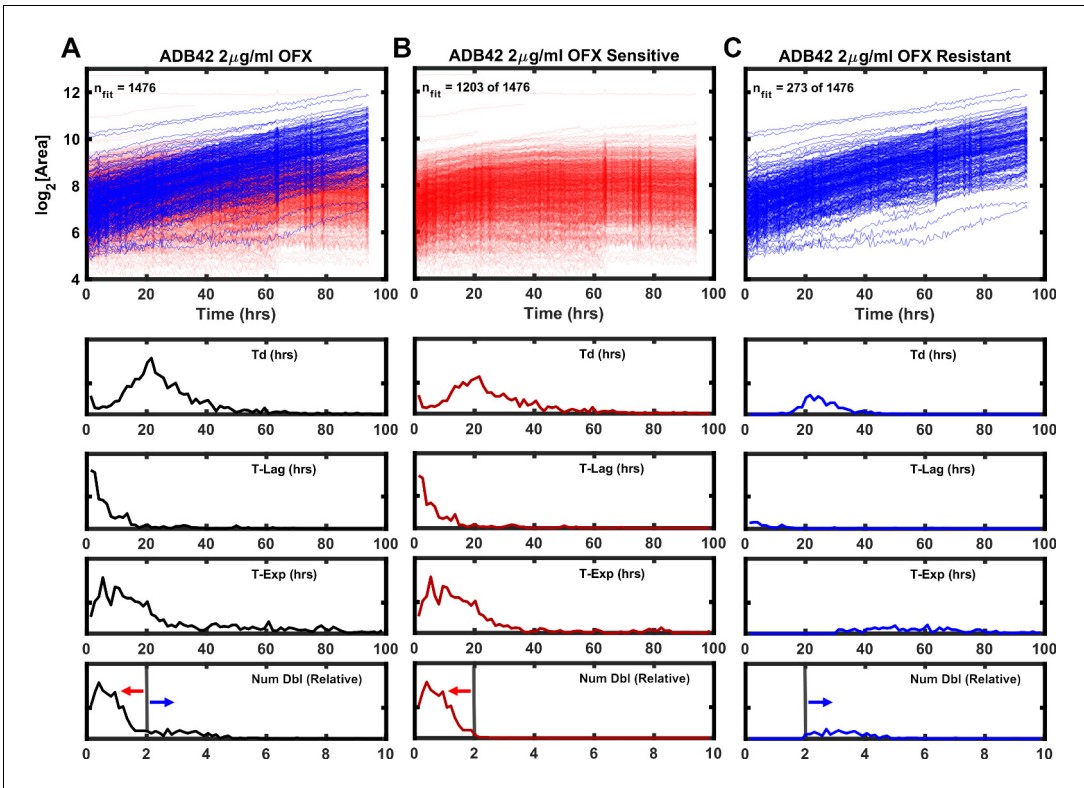

**Figure 7.** Growth kinetics generated by ODELAM for heteroresistant cultured clinical isolate ADB42. TOP (**A**) Growth curves of ADB42 growing at 2 μg/ml OFX. Of the 1476 CFUs in the heteroresistant culture, sensitive and resistant components were segregated by selecting those that doubled fewer than 2× (1203 CFUs; red traces) or more than 2× (273 CFUs; blue traces) (**B and C**) Traces of sensitive and resistant components presented separately for clarity. BOTTOM: Histograms of growth parameters derived from traces shown in **A** with sensitive and resistant components demarcated by the gray vertical lines.

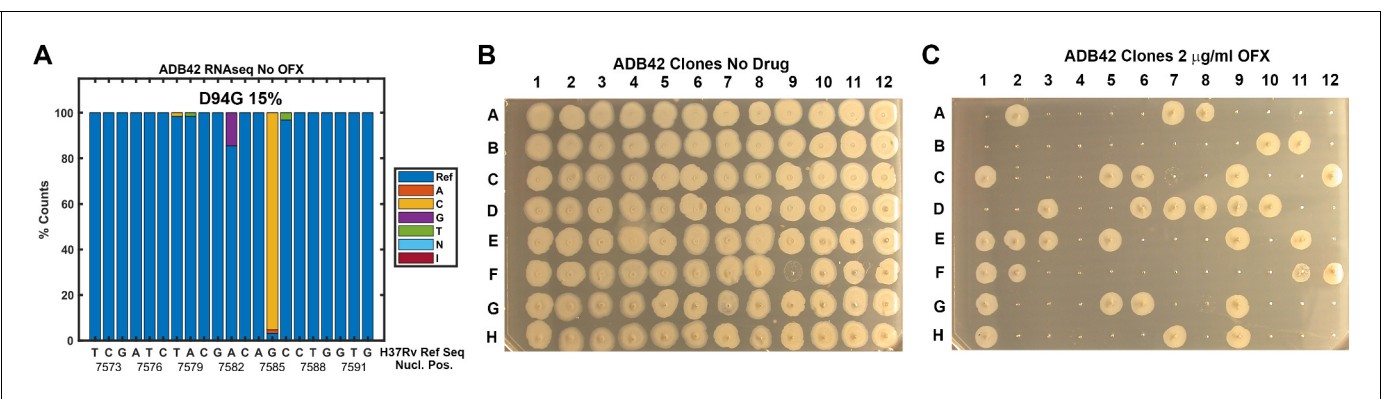

**Figure 8.** Assessment of ADB42 heteroresistance by spot assay. (**A**) Prevalence of the DNA gyrase II (Rv0006) D94G mutation before and after exposure to 32 μg/ml OFX for 24 hrs as detected by DNA sequencing. (**B and C**) Clones isolated from ADB42 grown on 7H10 media and 7H10 media with 2 μg/ml OFX for five days, before spotting on media lacking OFX (**B**) or containing OFX (**C**). 33 of 96 clones were observed to grow in the presence of OFX, indicating resistance to OFX.

**Table 2.** Ratios of resistant cells as calculated by application of the binomial distribution and experimental observations.

| Total number of sample reads (number of trials) | Number of D94G SNPs detected (number of success) p=r | Ratio of resistant cells to sensitive cells (probability of trial success) | 95% Confidence interval [Low]-[High] |
|---|---|---|---|
| 62 | 9 | 0.09–0.24 | [2, 9 - 9, 21] |
| Total number of colonies picked (number of trials) | Number of resistant colonies (number of success) p=r | | |
| 96 | 33 | 0.28–0.44 | [17,33 - 33,61] |
| | One or two cells contribute to colony p=1-(1 r)$^2$ | 0.14–0.26 | [17,33 - 33,61] |
| Total number of colonies tracked | Number that fit resistance criterion | Ratio of resistant cells to sensitive cells | |
| 963 | 237 | 0.2461 | |
| 473 | 114 | 0.2410 | |
| 1476 | 273 | 0.1850 | |
| 1043 | 244 | 0.2339 | |
| 359 | 101 | 0.2557 | |
| 561 | 157 | 0.2799 | |
| 573 | 146 | 0.2548 | |
| 459 | 123 | 0.2680 | |
| | Mean resistance ratio | 0.2456 | |

presence of OFX, exits exponential phase after ~20 hrs, ODELAM can detect drug sensitivity in ~30 hrs (*Figure 2*).

## Discussion

As the rates of antimicrobial-resistant infections rise, the need for fast and unbiased assays to detect and characterize antimicrobial-resistant bacterial infections are required. ODELAM is a time-lapse, microscopy-based assay that detects Mtb drug sensitivity within 30 hrs, offers an unparalleled view into the population dynamics of Mtb growth kinetics and can directly observe phenotypic heterogeneity. While commercial multiplexed automated digital microscope-based techniques are available to rapidly screen for antibiotic resistance (*Chantell, 2015*), here, using ODELAM we demonstrate the value of measuring growth kinetics on a single CFU level. With ODELAM. we observed dramatically different effects of OFX on the growth parameters of drug sensitive and drug resistant cultured clinical isolates of Mtb. These effects may offer insight into differing mechanisms of action for antimicrobials on Mtb or other organisms. ODELAM successfully discriminates between drug sensitive and drug resistant colonies in a mixed population. ODELAM also directly observes bacteriolysis and prolonged growth arrest, providing insight into the mechanisms of drug action. Based on this proof of principle, we suggest the investigation of additional cultured clinical isolates, additional Mtb drug classes, and advances in this technology to improve throughput and feasibility in resource limited settings.

### OFX effects on growth kinetics

Growth kinetics are a powerful indicator of genetic fitness. ODELAM's ability to monitor and evaluate growth kinetics of single CFUs from the point of plating through a time course of 100 hrs makes obtaining growth parameters and assessing drug sensitivity of multiple strains rapid and facile. Population growth kinetics under drug pressure illuminates the phenotypic differences between Mtb strains such as observed for the laboratory strain H37Rv, and an OFX resistant cultured clinical isolate TRS10 (*Figure 4*).

These two strains displayed very different phenotypic responses to OFX (*Figures 2*, *3* and *4*). H37Rv arrested growth at its MIC of 0.5 µg/ml as detected by departure from the exponential phase of growth. By contrast TRS10 grew continuously over a 96 hrs time period at its MIC OFX

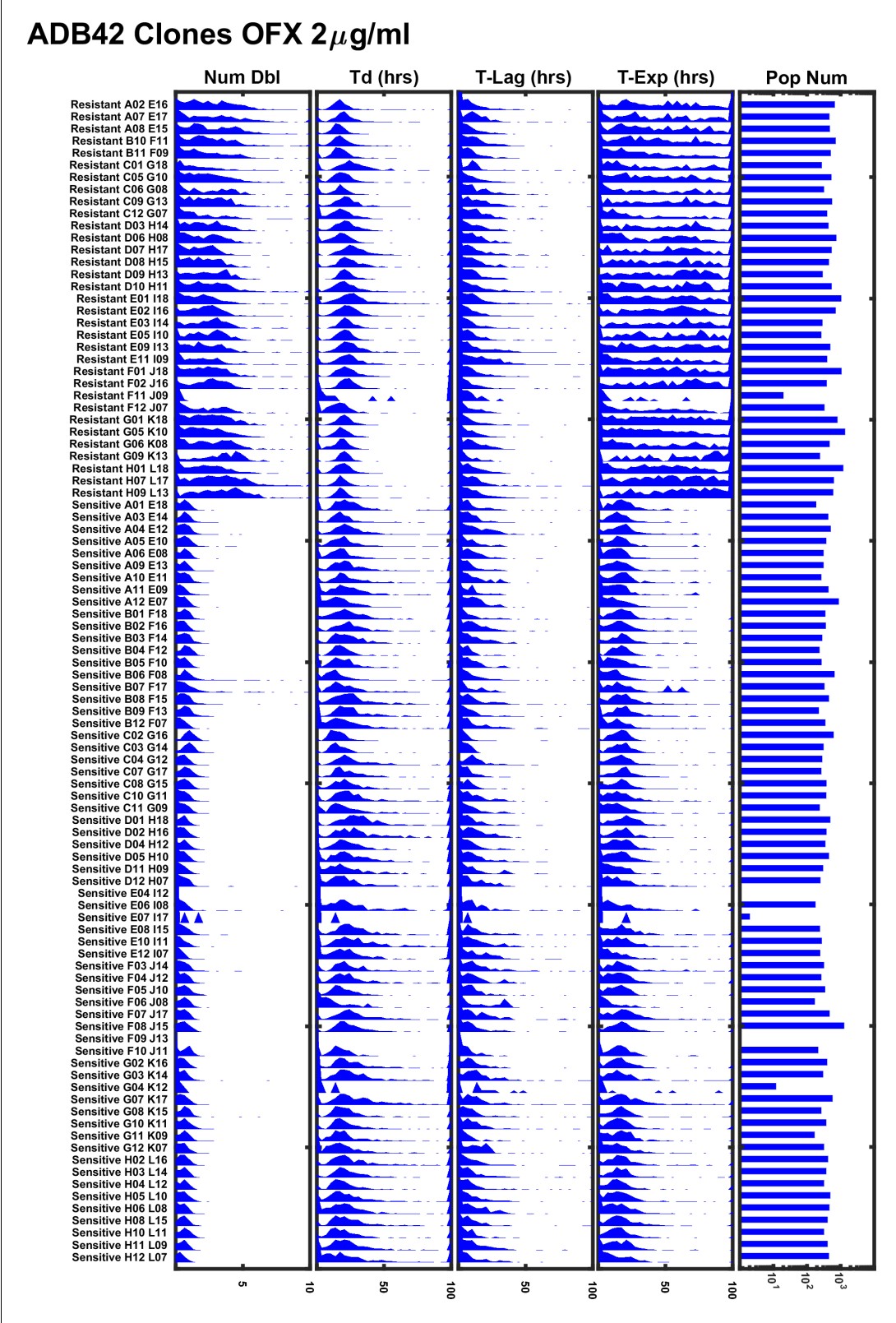

**Figure 9.** ODELAM experiment summary of 96 individual clones of ADB42 grown on 2 µg/ml OFX. Sensitive and resistant clones of ADB42 segregate according to their number of doublings (Num Dbl) and time in exponential phase (T-Exp). Sub-populations in the ADB42 clones were not observed.

**Table 3.** Sequence data *gyrA* SNPs in OFX sensitive and resistant clones.

| | | QRDR | QRDR | |
| --- | --- | --- | --- | --- |
| Position in the gyrase protein | 21 | 94 | 95 | 247 |
| H37Rv | E | D | S | G |
| TRS10 | Q | **G** | T | S |
| ADB42 OFX Sensitive | Q | D | T | G |
| ADB42 OFX Resistant | Q | **G** | T | G |

concentration of 16 µg/ml. Before growth arrest at its MIC, the doubling rate of H37Rv slowed slightly, whereas the doubling rate for TRS10 slowed considerably and at a population level exhibited a broad spread of doubling times. This suggests that the mechanisms of action for OFX toxicity are different between the two strains. As TRS10 contains the canonical D94G mutation in gyrase A conferring OFX resistance, we speculate that the growth phenotypes observed are due to the decreased affinity of OFX to the mutant DNA gyrase compared to wild-type, requiring a higher concentration of OFX to inhibit the enzyme. Thus, as the concentration of OFX is increased, off-target effects begin to manifest, which alter growth in a pleotropic manner resulting in the observed broadening of the doubling time of the population (*Figure 3*). By contrast, inhibition of DNA gyrase in H37Rv likely results in the accumulation of replication and transcription intermediates, leading to the more uniform population distribution observed.

Growth kinetics of Mtb under fluoroquinolone antibiotic pressure have not been previously reported with the level of detail presented here. Fluoroquinolones such as OFX bind to gyrase and trap the enzyme-DNA complex in an intermediate state (*Willmott et al., 1994*). The trapped complex can stall RNA transcription and prevent progression of the replication fork (*Willmott et al., 1994*; *Gore et al., 2006*; *Shea and Hiasa, 1999*; *Manjunatha et al., 2002*; *Hiasa et al., 1996*). In the presence of OFX, double stranded breaks in the DNA template free the RNA transcription complex and accumulate over time contributing to toxicity and cell death (*Willmott et al., 1994*; *Hiasa et al., 1996*). The time of exposure to OFX is thus directly linked to toxicity. Accordingly, even at OFX concentrations above the MIC (0.5 µg/ml), H37Rv continues to grow before they arrest synchronously after ~20 hrs (*Figure 2A*). We interpret this to mean that most cells remain metabolically

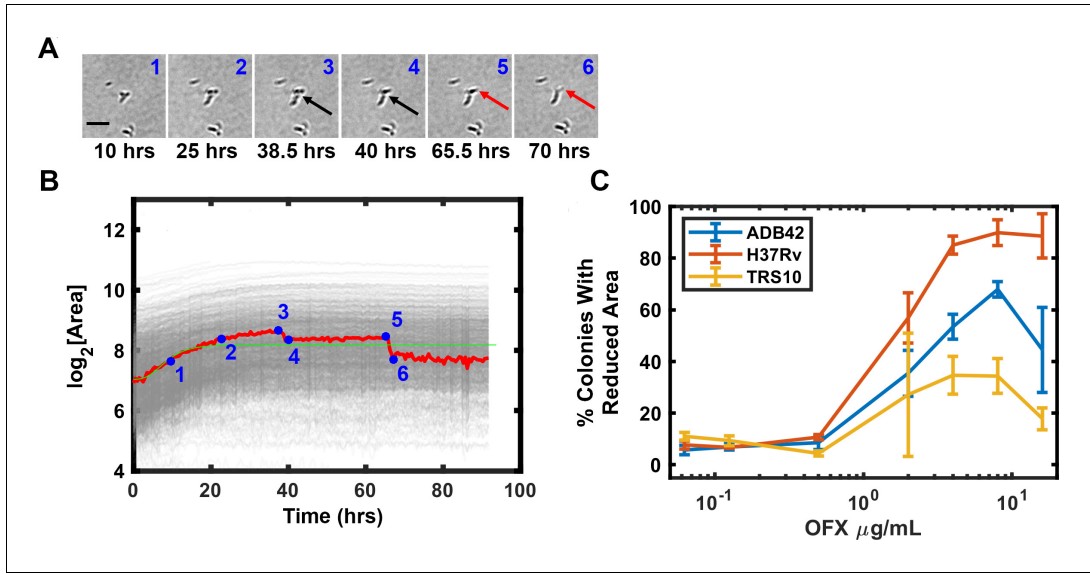

**Figure 10.** OFX induced lysis of Mtb bacilli. Some CFUs were observed to lose area after exposure to OFX, which is attributable cell lysis indicated by arrows (**A**). Tracking of images in A and plotting CFUs area over time detects lysis events as a reduction in CFU area (**B**). The percentage of colonies that lost area is plotted for each strain against drug concentration (**C**).

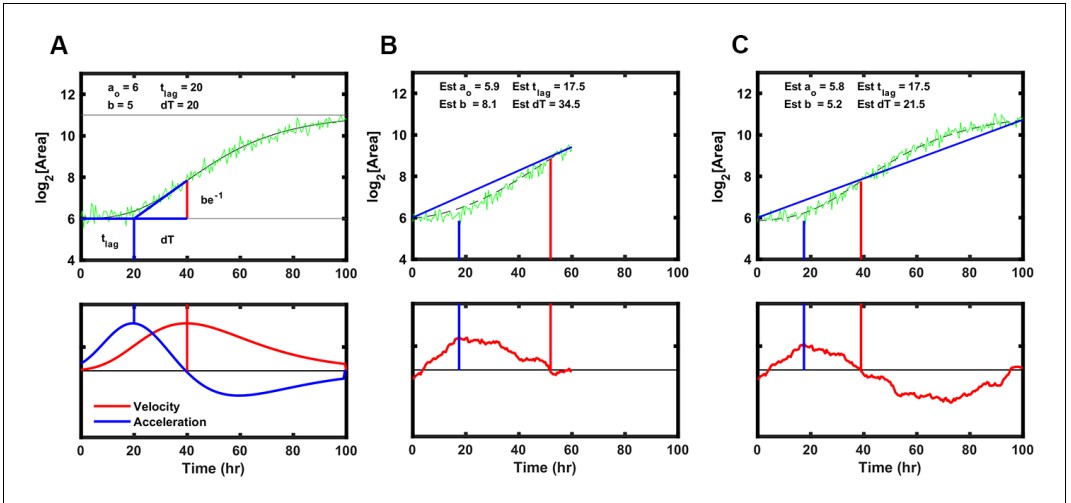

**Figure 11.** Strategy to fit Gompertz function parameters on a simulated growth curve with noise added. (A) Illustration of the parameters used to fit the Gompertz function as described in Materials and methods. Note the value dT, the time between T-Lag and the maximum growth velocity, is used for estimating the growth curve but T-Exp is reported. T-Exp is the time between the maximum and minimum acceleration or twice the value dT. (B) Initial estimates of each parameter were generated from a line connecting the first and last data points in a truncated simulated growth curve, which represents the time of experimental data acquisition. Bottom panel shows the differences between the line generated and the simulation. The initial estimate for T-Lag is determined by the maximum difference between the line generated and the simulation (blue vertical lines). T-Exp (or 2× dT) is determined where the absolute difference between the line and the simulated data is minimized (red vertical line). (C) Same as B with longer simulation time, showing initial estimates are similar.

active and synthesize protein until drug-induced damage accumulates and growth stops (*Wang et al., 2009*; *Da Re et al., 2009*).

## ODELAM can directly observe heteroresistant subpopulations

Standard microwell plate MIC assays and indicator tube assays are sensitive and robust methods for detecting antimicrobial resistant Mtb; however, these bulk assays give little information on the presence of sub-populations that differ in their sensitivity to drugs (*Lawson et al., 2013*). ODELAM directly observes and quantifies such heteroresistance in a mixed culture, segregating populations according to their growth kinetics.

As ODELAM directly observes growth kinetics and transitions to stasis, it can differentiate drug sensitivity with precision not available in bulk assays that rely on optical density or fluorescent growth reporters (*Kontos et al., 2004*; *Lawson et al., 2013*; *Garrigó et al., 2007*; *Kim, 2005*). This was validated with an artificially mixed population and was used to detect heteroresistance in a cultured clinical isolate, ADB42. In the mixed population ODELAM discriminates individual CFUs with a sensitivity of roughly 1 in 1000. This level of precision is dependent on the number of CFUs observed and improvements in microscope optics, computation, and media preparation could increase this sensitivity by orders of magnitude. Thus, this approach could observe the rare but critical events that lead to development of drug resistance including individual persistent cells that go on to acquire drug resistance through mutation (*Cohen et al., 2013*).

In the ADB42 isolate, OFX resistant cells were segregated by their growth kinetics in the presence of OFX. While DNA sequencing of this isolate failed to detect a resistant mutation, combining ODELAM with further DNA sequencing identified a D94G mutation in *gyrA* that we interpret conferred OFX resistance to a subpopulation in this cultured clinical isolate. These observations are consistent with other measurements of strains with *gyrA* mutations (*Eilertson et al., 2016*). Future advances of ODELAM that include real time monitoring, mapping and picking of single colonies will enable isolation and characterization of heterogenous populations based on their observed phenotypes.

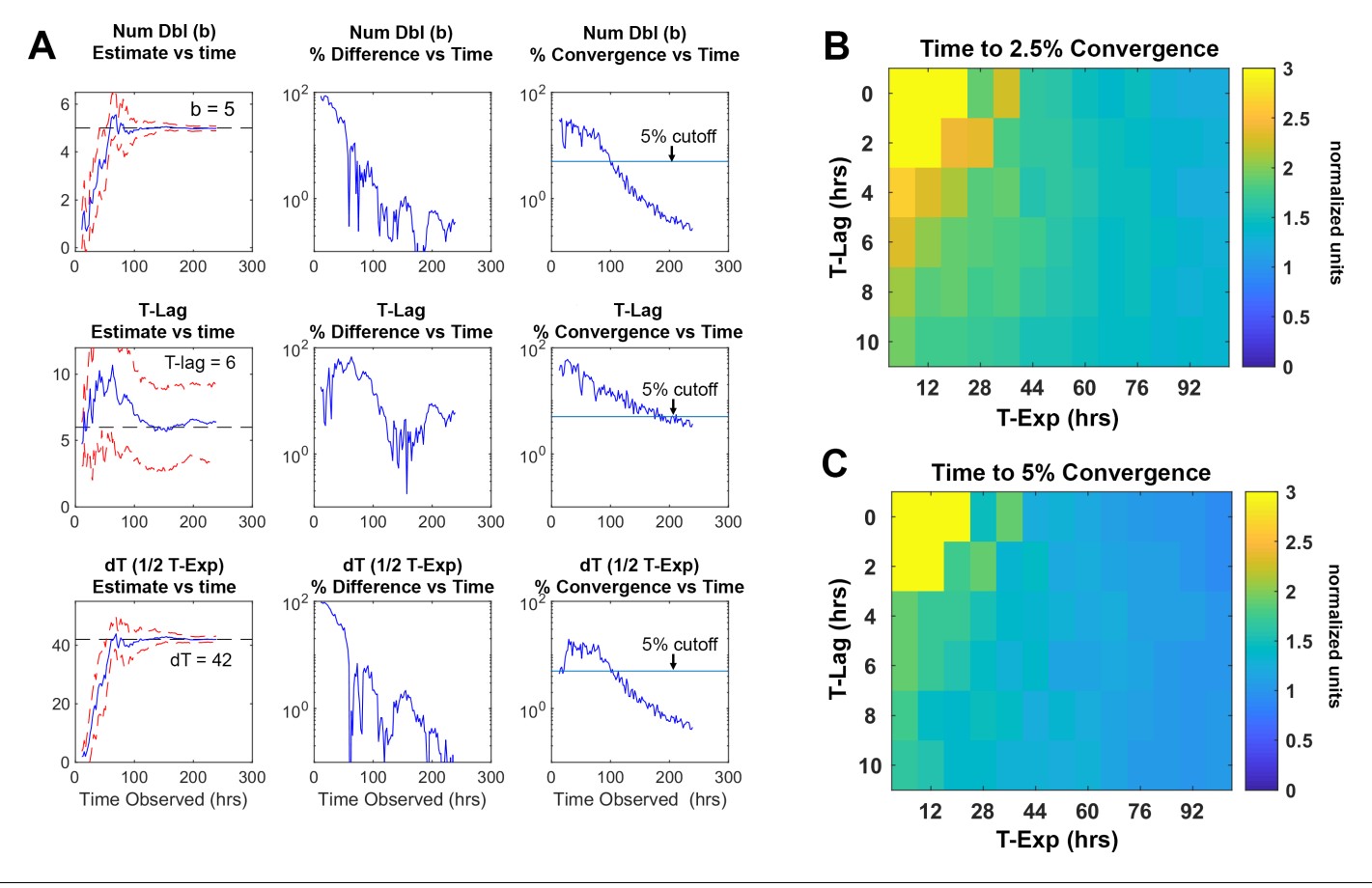

**Figure 12.** Time required for reliably determining the kinetic parameters in silico. (**A**) Column 1, the difference between the true parameter, in this case Num Dbl = 5, T-Lag = 6, and dT = 42, and the estimated parameter are plotted as more of the growth curve is observed over time. As more timepoints of the growth curve are included or "observed" in the fit algorithm, the estimated parameters approach their true values. The red dashed lines are 1 standard deviation of 25 growth curves each with random noise added. Column 2, the percent difference between the true value and the estimated value are unstable and do not decrease consistently as more time points are included in the estimation. Column 3, percent convergence given by the difference between two consecutive time point estimations and divided by the true parameter value and decreases steadily over time. The blue line represents a convergence threshold of 5% between consecutive estimations and this threshold is met when all parameters fall below the convergence threshold. (**B–C**) Heat maps showing the ratio of time needed for the *T-Exp* parameter to converge to a given precision, $T_o$ divided by the total time for the growth curve to begin to plateau *T-Lag + T-Exp*. The ratio is given explicitly as $\frac{T_o}{T_{lag} + T_{exp}}$. The heatmaps indicate that for given lag times and time in exponential phase the parameters converge after the growth curve begins to plateau. This means that for a drug that causes growth cessation at 20 hours, a measurement of roughly 1.5× that, or 30 hrs of observation, is required for the *T-Exp* parameter to converge to less 5% precision.

## Observations of cytostatic and cytotoxic phases of drug-microbe interactions

ODELAM has the ability to directly observe cytotoxicity and the impact of drugs through different phases of growth. Remarkably, for concentrations of OFX up to and including 16 times the MIC, there was no measurable effect on H37Rv's growth rate until growth ceased. By contrast OFX had a measurable effect on growth rate of TRS10 well below its MIC. We interpret these differences to reflect the differential accumulation of cytotoxic molecular intermediates due to differences in the ability of the drug to bind the different variants of gyrase. Nevertheless, ODELAM detected lysis of individual cells from each strain we analyzed in the presence of OFX at concentrations of 2 μg/ml and above (*Figure 10*), providing a visualization of the onset of cytotoxicity. These measurements provide better assessments of drug action by resolving growth, cytostatic and cytotoxic states on a single cell level.

## Conclusions

ODELAM is a powerful time-lapse imaging technique for rapidly evaluating biological growth phenotypes in Mtb. The current methods integrate with common automated microscope equipment and are amenable to a Biosafety Level 3 environment. ODELAM directly observed heteroresistance in an Mtb cultured clinical isolate and rapidly detected drug sensitivity in as little as 30 hrs. ODELAM is not restricted to Mtb and can be adapted to any colony forming microorganism. Insights from such studies promise an expanded view into the understanding of different mechanisms of antimicrobial resistance and drug action in Mtb and other important human pathogens.

# Materials and methods

**Key resources table**

| Reagent type (species) or resource | Designation | Source or reference | Identifiers | Additional information |
|---|---|---|---|---|
| Strain, strain background (*Mycobacterium tuberculosis*) | H37Rv | ATCC | ATCC −27294 | |
| Strain, strain background (*Mycobacterium tuberculosis*) | TRS10 | PMID:24687490 | 10 | |
| Strain, strain background (*Mycobacterium tuberculosis*) | ADB42 | PMID:24687490 | 42 | |
| Sequence-based reagent | Primer a for QRDR amplification | IDT | gyrA_1Fw | CCTGCGTTCG ATTGCAAACG |
| Sequence-based reagent | Primer b for QRDR amplification | IDT | gyrA_1Rv | CGTGGTTGACC TGATACGG |
| Commercial assay or kit | RiboZero rRNA removal (bacterial) | Illumina Inc | | (discontinued) |
| Commercial assay or kit | AMPure XP | Agencourt Bioscience Corporation | Beckman Coulter A63881 | |
| Commercial assay or kit | NEBNext Ultra RNA Library Prep Kit for Illumina | New England Biolabs | E7530S | |
| Commercial assay or kit | NEBNext Multiplex Oligos for Illumina | New England Biolabs | Dual Index Primers Set 1 E7335S | |
| Commercial assay or kit | Kapa qPCR quantification kit | Roche | KK4824 | |
| Commercial assay or kit | Illumina NextSeq 500 High Output v2 Kit | Illumina Inc | | |
| Commercial assay or kit | MagJet Genomic DNA kit | Thermo Fisher Scientific | K2722 | |
| Commercial assay or kit | Lysing Matrix B | MP Biomedicals | | |
| Chemical compound, drug | Ofloxacin | Sigma | O8757-1G | |

*Continued on next page*

*Continued*

| Reagent type (species) or resource | Designation | Source or reference | Identifiers | Additional information |
|---|---|---|---|---|
| Software, algorithm | DuffNGS | PMID:21317536 | | https://sourceforge.net/projects/duffyrnaseq/ |
| Software, algorithm | Bowtie 2 | PMID:22388286 | | |
| Software, algorithm | MicroManager v1.4 | PMID:25606571 | | https://micro-manager.org/ |
| Software, algorithm | MATLAB | Mathworks | | www.mathworks.com |
| Software, algorithm | ODELAY-ODELAM | This manuscript | | https://github.com/AitchisonLab/ |

## Mtb strains and culture methods

*Mycobacterium tuberculosis* was cultured as follows. Clinical isolates (*Eilertson et al., 2016*) were thawed and grown at 37 °C to an optical density of $OD_{600}$ 1.5 in 10 ml Middlebrook 7H9 media supplemented with glycerol, OADC supplement and 0.05% (v/v) Tween-80 (7H9-GOT) in 50 ml conical screw cap Falcon tubes. The samples were then stored at an $OD_{600}$ of 1 in 0.5 ml aliquots of 15% (v/v) glycerol and frozen at −80 °C until needed. For each experiment, a 0.5 ml aliquot was thawed and 9.5 ml of 7H9-GOT added. The culture was grown for 2–3 days until the $OD_{600}$ was 0.3–0.5. The sample was diluted to 0.05–0.10 $OD_{600}$ and grown for an additional 2 days to allow the culture to become well established in exponential growth phase. Finally, the cultures were diluted to 0.045 $OD_{600}$ for spotting with ODELAM. Three strains were investigated in this study: H37Rv (laboratory standard and OFX sensitive), TRS10 (OFX resistant cultured clinical isolate) and ADB42 (OFX hetero-resistant cultured clinical isolate). Strain names do not reflect their patient sources.

## Determination of the Minimum Inhibitory Concentration (MIC)

M.*M. tuberculosis* cultures were grown in 7H9-GOT medium at 37 °C to optical density of $OD_{600}$ 0.3–0.5. The cultures were then diluted to 0.05 $OD_{600}$ in fresh 7H9-GOT. A single 96-well plate was used for each strain. A 2-fold serial dilution of ofloxacin (0.015, 0.031, 0.062, 0.125, 0.25, 0.5, 1, 2, 4, 8, 16, 32, 64 μg/ml) was generated. Strains growing without drug were used as a 100% control and additional 1:100 dilution of each strain (to final $OD_{600}$ 0.0005) in the drug-free medium were used as 1% controls. Wells containing only medium were used to measure background fluorescence. The plates were incubated at 37 °C for 7 days. After incubation, viable Mtb cells were quantified using two methods, luminescence by applying the BacTiter Glo Microbial Cell Viability Assay reagent (Promega, Madison, WI, USA) and fluorescence by applying alamarBlue reagent (Bio-Rad Laboratories, Hercules, CA, USA). A volume of BacTiter-Glo reagent equal to the volume of cell culture medium present (20 μl) was added to each well. After 20 min of incubation at room temperature, luminescence was measured using the Omega plate reader. AlamarBlue was added in an amount equal to 10% of culture volume to each well of the 96-well plate. Plates were incubated at 37 °C for 9 hrs and the fluorescence was measured using an Omega plate reader, with an excitation wavelength at 544 nm and emission at 590 nm. MIC measurements were performed in three technical and two biological replicates. MIC values were defined as the lowest OFX concentration that inhibited growth compared to growth of the 1% control.

## Mtb cloning and sequencing

Clinical isolates ADB42 and TRS10 were grown as previously described (*Eilertson et al., 2014*; *Eilertson et al., 2016*). After the culture reached an $OD_{600}$ 0.5 the culture was diluted to $10^5$, $10^6$, and $10^7$ CFUs/ml and plated on a petri dish containing 7H10 media. Single colonies were then picked with a pipette tip and subcultured in a 100 μl of 7H9-GOT media in a 96 well plate. After growth to an $OD_{600}$ of ~0.2 per well, cultures were diluted 10× and cultured at 37 °C for an additional 3 days. Then OFX resistance was evaluated by spotting 2 μl of clonal cultures onto 7H9 media with 0 or 2 μg/ml OFX. After two weeks of growth, agar plates were evaluated and photographed

to determine resistant and sensitive clones. Additionally, the 96 clones were evaluated with ODE-LAM by spotting onto 7H9-GO with 2 µg/ml OFX and growth was recorded for 96 hrs.

The quinolone resistance-determining region (QRDR) of gyrase A was amplified by PCR using the following method. ADB42, TRS10, and H37Rv were grown to an $OD_{600}$ of 0.5 in 10 ml cultures. The cultures were pelleted by centrifugation and resuspended in 0.53 ml of TE buffer. Cells were mechanically lysed three times for 30 s with 0.1 mm silica beads (Lysing Matrix B, MP Biomedicals). The samples were then pelleted by centrifugation and the supernatants were transferred into new tubes and heated for 30 min at 105 °C. DNA was purified using a MagJet Genomic DNA kit (Thermo Fisher Scientific) according to the manufacturer's instructions. DNA fragments for sequencing were amplified by PCR using a mix of 1 ng of genomic DNA, 5 pmol of each primer (gyrA_1Fw and gyrA_1Rv), 200 µM dNTPs, 1× Prime Star buffer and Prime Star polymerase. The primers used are shown in Table 4. PCR products were purified using a NucleoSpin Gel and PCR clean up kit (Macherey Nagel). Sanger sequencing of PCR products was performed by GENEWIZ (Seattle, WA, USA).

## RNA isolation and mtb transcriptome sequencing and analysis

RNA was isolated from cultures as described previously (*Sherman et al., 2001*). Briefly, cell pellets resuspended in Trizol were transferred to a tube containing Lysing Matrix B (MP Biomedicals) and disrupted by homogenization at maximum speed for 30 s in a FastPrep 120 homogenizer (QBiogene) three times, with cooling on ice between cycles. The homogenate was centrifuged in a micro-centrifuge at maximum speed for 1 min and the supernatant was transferred to a tube containing 300 µl chloroform and Heavy Phase Lock Gel (Eppendorf). Following mixing by inversion, the samples were centrifuged in a microcentrifuge tube at maximum speed for 5 min. RNA in the aqueous phase was then precipitated with 300 µl isopropanol and 300 µl high salt solution (0.8 M Na citrate, 1.2 M NaCl). RNA was purified using a RNeasy kit following the manufacturer's recommendations (Qiagen) with one on-column DNase treatment (Qiagen). Total RNA yield was quantified using a Nanodrop (Thermo Scientific).

To enrich the mRNA, ribosomal RNA was depleted from samples using a RiboZero rRNA removal (bacteria) magnetic kit (Illumina Inc, San Diego, CA). The products of this reaction were prepared for Illumina sequencing using the NEBNext Ultra RNA Library Prep Kit for Illumina (New England Biolabs, Ipswich, MA) according to manufacturer's instructions, and using the AMPure XP reagent (Agencourt Bioscience Corporation, Beverly, MA) for size selection and cleanup of adaptor-ligated DNA. NEBNext Multiplex Oligos for Illumina (Dual Index Primers Set 1) were used to barcode the DNA libraries associated with each replicate and enable multiplexing of 96 libraries per sequencing run. The prepared libraries were quantified using a Kapa qPCR quantification kit, and were sequenced at the University of Washington Northwest Genomics Center with the Illumina NextSeq 500 High Output v2 Kit (Illumina Inc, San Diego, CA). The sequencing generated an average of 75 million base-pair paired-end raw read counts per library.

Raw FASTQ read data were processed using in-house R package DuffyNGS, as originally described (*Vignali et al., 2011*). Briefly, raw reads pass through a 3 stage alignment pipeline: 1) a pre-alignment stage to filter out unwanted transcripts, such as ribosomal RNA; 2) a main genomic alignment stage against the genome(s) of interest; 3) a splice junction alignment stage against an index of standard and alternative exon splice junctions. All alignments were performed with Bowtie2, using the command line option '–very-sensitive' (*Langmead and Salzberg, 2012*). BAM files from stages 2 and 3 were combined into read depth wiggle tracks that record both uniquely mapped and multiply mapped reads to each of the forward and reverse strands of the genome(s) at single nucleotide resolution. Multiply mapped reads were pro-rated over all highest quality aligned locations. Gene transcript abundance was then measured by summing total reads landing inside annotated gene boundaries or exon boundaries, expressed as both RPKM and raw read counts (*Wold and Myers, 2008*). Two stringencies of gene abundance were provided, using all aligned reads or by using just uniquely aligned reads.

## Preparation of ODELAM agarose plates

A Ninjaflex 2 mm thick gasket was 3D printed onto a 50 mm × 75 mm × 1 mm glass slide (VWR) using a Makerbot 2× printer. Following printing, the slides were cleaned of particles and fibers using

lab tape. An additional slide was cleaned with ethanol and placed over the gasket so the space between the slides defined by the gasket could be filled with molten agarose media. The assembled agar mold was held together with binder clips.

Agarose was prepared as described previously (*Herricks et al., 2017*). Bulk agarose was prepared by dissolving 2 g of agarose in 150 g of 18 MΩ $H_2O$. The solution was microwaved in 15 s intervals for about 3 min or until the agarose boiled and all agar particles were dissolved by visual inspection. The 3.1 g of molten agarose was then aliquoted into 15 ml Falcon tubes and stored at 4 ° C. On the morning of an experiment, 0.4 ml of 10× 7H9 G media, 0.2 ml of sterile 18 MΩ $H_2O$ were added to five previously prepared 3 g aliquots of agarose tubes. The solution was boiled for 8 min in a covered boiling flask. After fully melting the agarose, the solution was placed into a 45 °C stirred bath to cool for 2–10 min. Once cooled, 0.4 ml of Middlebrook OADC and 4 μl of 1000× concentration of drug or drug vehicle were added to make a total of 4 ml of 7H9 glycerol oleate (7H9-GO) agarose media. The media was then injected into the single or each of the 5 glass-gasket-glass chambers using a 20-gauge syringe needle. The agarose was allowed to set for about 30 min and one glass slide was removed to expose the agar surface. The agarose slide was then placed in a sterile pipette tip box and stored at room temperature for about 2 hrs before spotting Mtb cultures.

## ODELAM Time-Lapse microscopy and image analysis

The agar plate was assembled into the ODELAM growth chamber and 1 μL of 0.045 $OD_{600}$ Mtb culture spotted on the agar plate using a fabricated tool for spotting cells in fixed locations corresponding to positions E06 through L19 of a 384 well plate defining the center of each region of interest for imaging. The chamber was assembled, and electrical contacts placed on an Indium Tin-Oxide (ITO) coated cover-slide. The ITO slide was resistively heated with 10 V and 0.2 A of current to minimize condensation above the agar. The spotted CFUs were imaged using on a Nikon TiE microscope equipped with an In Vitro Scientific incubator maintaining a temperature of 37 °C. Images were recorded using a 20× 0.45 NA long working distance lens with the correction collar set to 1.3 mm. This collar setting was required to get a monomodal focus function as calculated using the Laplacian Variance Method (*Pertuz et al., 2013*). A Photometrix CoolSnap $ES^2$ camera recorded brightfield images. The microscope was controlled by MATLAB Graphical User Interface software using the MicroManager Core API (*Edelstein et al., 2014*). At each location where culture was spotted, the microscope tiled a 3 × 3 array with 20% overlap. Each spot was imaged every 30 min for up to 120 hrs. Analysis of the images was performed as described in *Herricks et al., 2017*. Briefly, the resulting images were stitched, binarized with a threshold and colonies were identified from the binarized images. Objects were tracked using a MATLAB Image processing pipeline. If objects merged as indicated by two previous separate centroids appearing in a continuous colony patch, the track was no longer recorded. If a track was not observed for a minimum of 30 hrs we did not include that colony's statistics in the analysis. Results were then quantified and plotted using MATLAB. ODELAM is optimized to track between approximately 50 to 1500 individual cells and CFUs for 96 hrs. The lower limit of about 50 CFUs per field of view is required to have sufficient contrast to focus and stitch the images.

## Extraction of growth parameters

Growth curves of $\log_2$ area over time were fit to a parameterized version of the Gompertz function:

$$A(t) = a_0 + be^{-e^{\left[\frac{\log\left(\frac{3+\sqrt{5}}{2}\right)}{dT}\left(dT+t_{lag}-t\right)\right]}}$$

Where $a_0$ is the initial area of the cells, $b$ is the number of doublings the colony undergoes until quiescence, $t_{lag}$ is the time before the onset of exponential growth and $dT$ is ½ T-Exp, the time the cells grow exponentially. Estimation of growth parameters was performed using the following strategy: The parameters of the Gompertz function given by lag time $t_{lag}$ and $dT$ correspond to the maximum of the function's 1st and 2nd derivatives, respectively (*Figure 11A*). To determine these values, we used the MATLAB function fmincon which uses a nonlinear least squares approach to estimate the parameters by minimizing the residuals of the observed data and the parameterized Gompertz function. Initial values for the growth parameters estimation routine were determined using either a

geometric approximation of initial parameters or a course grid search. The geometric approximation of initial values used the following strategy: A line between the first and last points of the measured growth curve (*Figure 11B*, blue line) was defined such that the maximum difference between that line and the smoothed growth curve approximated the growth curve's second derivative (*Figure 11B*). The maximum of this approximated second derivative estimates lag time and is used as the initial parameter for determining lag time in the fmincon function. Likewise, the location where this second derivative crosses zero estimates the maximum velocity and is used as the initial parameter for determining the maximum velocity with the fmincon function (*Figure 11B and C*). The value for $a_0$ is estimated from averaging the area of the first 5 data points and $b$ is estimated by the area of the growth curve evaluated at the maximum velocity minus $a_0$. A course grid optimization routine was used if the geometric approximation failed (*Herricks et al., 2017*). Doubling time is derived from the two parameters $b$ and $dT$ by:

$$T_d = e^1 b \frac{dT}{log\left(\frac{3+\sqrt{5}}{2}\right)}$$

We also applied the constraints that $t_{lag} + dT$ must be less than or equal to the total time observed, which restricts growth parameters to values that occur during observation and we only included colonies that were tracked for greater than 30 hrs, and showed a $log_2$ growth greater than 0.1. While the area measurements have noise associated with focus accuracy, the camera readout, and illumination instability, we image at a high temporal rate such that repeated measurements minimize the effect of noise on fitting the data. All data were plotted using MATLAB software.

## Statistical analysis of ODELAM data

Effect size was measured by pooling replicate measurements into a single population. The resulting population distributions were compared using the 2-sample Kolmogorov-Smirnov test against a no-drug control pool. The KS test statistic for each comparison was then plotted to show the trend in OFX concentration vs. effect size. The binomial distribution was used to estimate the percentage of resistant cells in ADB42. For clarity, the binomial probability distribution function is given by:

$$f(k,n,p) = \frac{n!}{k!(n-k)!}p^k(1-p)^k$$

where *n* is the number of resistant colonies or SNP reads, *k* is the total number of colonies picked or the total number of reads at the SNP locus and *p* is the fraction of resistant CFUs in the culture. The binomial cumulative distribution function was calculated using the MATLAB function cdf. In this way, we modeled the number of colonies that appeared resistant or the number of whole genome sequencing reads that contain the D94G mutation in Rv0006 as a function of the percentage of cells in the culture that are OFX resistant. The probability of success (*p*) is given by the percentage of OFX resistant cells which is estimated by the number of successful trials (given by the number of SNP reads or the number of resistant colonies) and is bounded by the 95% confidence interval that still contains the actual number of WGS reads or resistant colonies (i.e. number of successes) observed.

## Additional information

### Funding

| Funder | Grant reference number | Author |
| --- | --- | --- |
| National Institutes of Health | U19 AI135976 | David R Sherman<br>John D Aitchison |
| National Institutes of Health | U19 AI111276 | John D Aitchison |
| National Institutes of Health | R01 AI141953 | John D Aitchison |
| National Institutes of Health | P41 GM109824 | John D Aitchison |
| National Institutes of Health | R01 AI063200 | Timothy R Sterling |

| National Institutes of Health | R56 AI118361 | Timothy R Sterling |

The funders had no role in study design, data collection and interpretation, or the decision to submit the work for publication.

## Author contributions

Thurston Herricks, Conceptualization, Data curation, Software, Formal analysis, Validation, Investigation, Visualization, Methodology, Writing - original draft, Writing - review and editing; Magdalena Donczew, Conceptualization, Resources, Data curation, Formal analysis, Validation, Investigation, Visualization, Methodology, Writing - original draft, Writing - review and editing; Fred D Mast, Conceptualization, Writing - original draft, Writing - review and editing; Tige Rustad, Conceptualization, Formal analysis, Investigation, Methodology; Robert Morrison, Software, Formal analysis, Writing - review and editing; Timothy R Sterling, Conceptualization, Resources, Funding acquisition, Writing - review and editing; David R Sherman, Conceptualization, Resources, Supervision, Funding acquisition, Validation, Investigation, Visualization, Methodology, Writing - original draft, Project administration, Writing - review and editing; John D Aitchison, Conceptualization, Resources, Formal analysis, Supervision, Funding acquisition, Validation, Investigation, Visualization, Methodology, Writing - original draft, Project administration, Writing - review and editing

## Author ORCIDs

Thurston Herricks (iD) https://orcid.org/0000-0002-0247-7967
Fred D Mast (iD) https://orcid.org/0000-0002-2177-6647
John D Aitchison (iD) https://orcid.org/0000-0002-9153-6497

## Decision letter and Author response

Decision letter https://doi.org/10.7554/eLife.56613.sa1
Author response https://doi.org/10.7554/eLife.56613.sa2

# Additional files

## Supplementary files

• Transparent reporting form

## Data availability

MATLAB data *.mat files and MATLAB *.m files utilized for generating figures in this submission are posted at Dryad. Additional source code has been made available at https://github.com/AitchisonLab/.

The following dataset was generated:

| Author(s) | Year | Dataset title | Dataset URL | Database and Identifier |
|---|---|---|---|---|
| Herricks T, Donczew M, Mast FD, Rustad T, Morrison R, Sterling TR, Sherman DR, Aitchison JD | 2020 | ODELAM: Rapid sequence-independent detection of drug resistance in clinical isolates of Mycobacterium tuberculosis | http://dx.doi.org/10.5061/dryad.b8gtht78q | Dryad Digital Repository, 10.5061/dryad.b8gtht78q |

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
