## [Decision Letter]

**Acceptance summary:**

Drug resistance is increasingly common in *Mycobacterium tuberculosis* infection, and a means to rapidly assess the level and nature of potential drug resistance is important for treatment. Current approaches for measuring antimicrobial resistance involve either genotypic assays or relatively slow phenotypic assays involving prolonged in vitro culture. This work proposes a rapid phenotypic assessment of drug resistance by combining microscopic assessment of colony growth with statistical analysis of colony growth rates. This has the potential also to rapidly identify resistant strains, as well as identify small levels of drug resistance in a heterogenous mixture of bacteria.

**Decision letter after peer review:**

Thank you for submitting your article "ODELAM: Rapid sequence-independent detection of drug resistance in clinical isolates of *Mycobacterium tuberculosis*" for consideration by *eLife*. Your article has been reviewed by three peer reviewers, and the evaluation has been overseen by a Reviewing Editor and Wendy Garrett as the Senior Editor. The following individual involved in review of your submission has agreed to reveal their identity: Andrew J Yates (Reviewer #3).

The reviewers have discussed the reviews with one another and the Reviewing Editor has drafted this decision to help you prepare a revised submission.

Summary:

The authors describe a new method ODELAM for rapid testing the drug sensitivity of Mtb to a range of antibiotics. The uniqueness of this method lies in the temporal analysis across the bacterial population at the single CFU level, this provides extremely impressive resolution across a complex, heterogeneous bacterial population. The paper contains experiments with spiked populations of sensitive and resistant bacteria as well as analysis of clinical isolates. Furthermore, they demonstrate that they can not only distinguish drug sensitive from drug resistant Mtb strains, but also strains with genotypically mixed resistant/sensitive populations. The work has been performed with care and rigor and is of considerable value to the field.

Essential revisions:

1) Using timelapse microscopy to speed up detection of resistance patterns in Mtb is logical and, as the authors demonstrate, gives resolution not only into drug resistance of Mtb infection, but also into the heterogeneity of responses to drug. Therefore, the premise of the work is strong, and the authors demonstrate technical prowess with acquiring and analysing the data. What the paper does not accomplish is its stated goal, of demonstrating that timelapse microscopy could be used to accelerate diagnosis. This is because while they use clinical isolates of Mtb, these are coming from lab culture in the same way as H37Rv, that is, they are grown to OD ~0.5 in Middlebrook 7H9 media, then diluted and plated for microscopy. This is very different from the sample used for diagnosis, which is a sputum sample. The sputum sample may have Mtb at very different concentrations between individuals, it may have contaminants that may be identified as Mtb by image analysis, the Mtb may have a much more variable lag time, etc. While these problems may be solvable, they are not solved in this work and so the feasibility of using this method for its stated purpose is unknown.

To address this, the authors should ideally provide evidence that the approach can be used on clinical samples. However, recognising the current difficulties in carrying out additional experimental work, the authors may alternatively choose to include a caveat that this has not been demonstrated on clinical samples (this needs to be clearly articulated throughout the manuscript in any section where claims to clinical utility are made).

2) The Introduction and referencing are weak in describing what was done previously in the field and what advance this work presents:

a) I could not find a "References" section using the find command.

b) A quick online check using "*Mycobacterium tuberculosis*timelapse" and "*Mycobacterium tuberculosis* image analysis" came up with up with Barr et al., 2016, Choi et al., 2016 and Hertog et al., 2010. These all seem to be using timelapse on Mtb microcolonies.

c) For completeness, authors should discus accredited technologies such as MGIT, which are more rapid than CFU.

In the Introduction the authors fail to mention the fluorescent and luminescent readouts that have been used by several labs for drug screening assays and to assess drug sensitivity, these provide a readout in a matter of days. While the bulk of these assays have been conducted at the population level there are some groups that have used high content imaging that has provided greater resolution across the bacterial population. I agree completely with the authors that the platform detailed here is far superior and has much greater resolution because the data are temporal and have single clone/colony resolution, nonetheless the authors do need to accurately present the current state of the field.

Thus, the authors must include better referencing of previous work to identify what has been previously tried by others, and what advantages/disadvantages their work brings compared to previous approaches.

---

## [Author Response]

Essential revisions:1) Using timelapse microscopy to speed up detection of resistance patterns in Mtb is logical and, as the authors demonstrate, gives resolution not only into drug resistance of Mtb infection, but also into the heterogeneity of responses to drug. Therefore, the premise of the work is strong, and the authors demonstrate technical prowess with acquiring and analysing the data. What the paper does not accomplish is its stated goal, of demonstrating that timelapse microscopy could be used to accelerate diagnosis. This is because while they use clinical isolates of Mtb, these are coming from lab culture in the same way as H37Rv, that is, they are grown to OD ~0.5 in Middlebrook 7H9 media, then diluted and plated for microscopy. This is very different from the sample used for diagnosis, which is a sputum sample. The sputum sample may have Mtb at very different concentrations between individuals, it may have contaminants that may be identified as Mtb by image analysis, the Mtb may have a much more variable lag time, etc. While these problems may be solvable, they are not solved in this work and so the feasibility of using this method for its stated purpose is unknown.To address this, the authors should ideally provide evidence that the approach can be used on clinical samples. However, recognising the current difficulties in carrying out additional experimental work, the authors may alternatively choose to include a caveat that this has not been demonstrated on clinical samples (this needs to be clearly articulated throughout the manuscript in any section where claims to clinical utility are made).

We agree with the reviewers that it is important to be able to rapidly diagnose Mtb from clinical samples and observing Mtb directly from clinical samples such as sputum is a work in progress. While we did test clinical isolates, as the reviewers point out, these were not clinical samples. To make this important distinction clear in the manuscript, we now refer to the clinical isolates as “cultured clinical isolates” throughout the text and changed the title accordingly. See the Abstract, and Introduction for examples. We have been actively working on adapting ODELAM for use with clinical samples and as suggested, will report the relevant conclusions in a preprint and, if appropriate, as a Research Advance in *eLife*.

2) The Introduction and referencing are weak in describing what was done previously in the field and what advance this work presents:a) I could not find a "References" section using the find command.

Thank you for pointing out the lack of a References section, which was somehow omitted during submission; we regret the mistake. Additional references requested by the reviewers are also included.

b) A quick online check using "Mycobacterium tuberculosis timelapse" and "Mycobacterium tuberculosis image analysis" came up with up with Barr et al., 2016, Choi et al., 2016 and Hertog et al., 2010. These all seem to be using timelapse on Mtb microcolonies.

We thank the reviewers for pointing us to these references. We now include them in addition to prior work, which directly influenced development of our platform (Introduction).

c) For completeness, authors should discus accredited technologies such as MGIT, which are more rapid than CFU.In the Introduction the authors fail to mention the fluorescent and luminescent readouts that have been used by several labs for drug screening assays and to assess drug sensitivity, these provide a readout in a matter of days. While the bulk of these assays have been conducted at the population level there are some groups that have used high content imaging that has provided greater resolution across the bacterial population. I agree completely with the authors that the platform detailed here is far superior and has much greater resolution because the data are temporal and have single clone/colony resolution, nonetheless the authors do need to accurately present the current state of the field.Thus, the authors must include better referencing of previous work to identify what has been previously tried by others, and what advantages/disadvantages their work brings compared to previous approaches.

We agree with the reviewers’ comments about providing an accurate current state of the field have expanded our Introduction to include these references and now specifically mention agar plate and indicator tube assays in a description of the current state of the field (Introduction and subsection “ODELAM can directly observe heteroresistant subpopulation”).